# Cancer Incidence and Risk of Multiple Cancers after Environmental Asbestos Exposure in Childhood—A Long-Term Register-Based Cohort Study

**DOI:** 10.3390/ijerph19010268

**Published:** 2021-12-27

**Authors:** Sofie Bünemann Dalsgaard, Else Toft Würtz, Johnni Hansen, Oluf Dimitri Røe, Øyvind Omland

**Affiliations:** 1Department of Clinical Medicine, Aalborg University, 9000 Aalborg, Denmark; 2Danish Ramazzini Centre, Department of Occupational and Environmental Medicine, Aalborg University Hospital, 9000 Aalborg, Denmark; ELSWUR@rm.dk (E.T.W.); omland@dcm.aau.dk (Ø.O.); 3Danish Ramazzini Centre, Department of Occupational Medicine, University Research Clinic, Goedstrup Hospital, 7400 Herning, Denmark; 4Danish Ramazzini Centre, Department of Occupational Medicine, Aarhus University Hospital, 8200 Aarhus, Denmark; 5Danish Cancer Society Research Center, 2100 Copenhagen, Denmark; Johnni@cancer.dk; 6Department of Oncology, Aalborg University Hospital, 9000 Aalborg, Denmark; olufdroe@yahoo.no; 7Department of Clinical and Molecular Medicine, Norwegian University of Science and Technology, 7491 Trondheim, Norway; 8Department of Oncology, Levanger Hospital, Nord-Trøndelag Hospital Trust, 7600 Levanger, Norway

**Keywords:** cancer, asbestos, registry study, mesothelioma, pharynx cancer, environmental exposure, childhood

## Abstract

Objectives: To examine the asbestos-associated cancer incidence and the risk of multiple cancers in former school children exposed to environmental asbestos in childhood. Methods: A cohort of 12,111 former school children, born 1940–1970, was established using 7th grade school records from four schools located at a distance of 100–750 m in the prevailing wind direction from a large asbestos-cement plant that operated from 1928 to 1984 in Aalborg, Denmark. Using the unique Danish personal identification number, we linked information on employments, relatives’ employments, date of cancer diagnosis, and type of cancer and vital status to data on cohortees extracted from the Supplementary Pension Fund Register (employment history), the Danish Cancer Registry, and the Danish Civil Registration System. We calculated standardized incidence rates (SIRs) for asbestos-associated cancers, all cancers, and multiple cancers using rates for a gender and five-year frequency-matched reference cohort. Results: The overall incidence of cancer was modestly increased for the school cohort (SIR 1.07, 95% confidence interval (CI) 1.02–1.12) compared with the reference cohort. This excess was driven primarily by a significantly increased SIR for malignant mesothelioma (SIR 8.77, 95% CI 6.38–12.05). Former school children who had combined childhood environmental and subsequent occupational exposure to asbestos had a significantly increased risk of lung cancer. Within this group, those with additional household exposure by a relative had a significantly increased SIR for cancer of the pharynx (SIR 4.24, 95% CI 1.59–11.29). We found no significant difference in the number of subjects diagnosed with multiple cancers between the two cohorts. Conclusions: Our study confirms the strong association between environmental asbestos exposure and malignant mesothelioma and suggests that environmental asbestos exposure in childhood may increase the overall cancer risk later in life.

## 1. Introduction

Health risks associated with asbestos exposure, including the risk of certain cancers, have been known for many years, but research has been devoted mainly to occupational asbestos exposure or exposure in adults in general, and rarely to asbestos exposure in childhood [1]. Asbestos continues to be used around the world. This poses a risk not only for those working with asbestos, but also for the general population, including children who are exposed to environmental asbestos [2,3,4,5,6,7]. The International Agency for Research on Cancer (IARC) has concluded that all forms of asbestos may cause mesothelioma and cancer of the lung, larynx, and ovary [8]. Furthermore, the IARC has found a positive association between asbestos exposure and cancer of the pharynx, stomach, colon, and rectum, though the epidemiological evidence for these cancers is limited. The primary routes of exposure to asbestos are inhalation and ingestion [8].

We have studied environmental asbestos exposure both in adults and children and found a significantly increased risk for mesothelioma development [9,10]. We also found some evidence for an increased risk of haematological cancers in children exposed to asbestos [11]. Only a few studies [12,13] have investigated to which extent prior asbestos exposure contributes to the increasing incidence of multiple primary cancers, seen in recent decades with the advent of better diagnostic techniques, increasing longevity, and improved survival in cancer patients [14]. One study proposed that a gene/environment interaction involving BAP1, environmental exposure to asbestos, and UV irradiation played a role in a family with a high incidence of multiple primary cancers [13]. In another study, five patients with multiple primary tumors were found to have had a history of occupational asbestos exposure [12].

Denmark had one factory which was a producer of asbestos-containing products, until an asbestos ban was introduced in the 1980s. The Danish asbestos cement plant (Dansk Eternit Fabrik A/S) was located in Aalborg city in Northern Denmark, where production of asbestos cement products took place from 1928 until 1988. A total of approximately 620,000 tons of asbestos (89% chrysotile) was consumed during the production period [15].

The cancer incidence among the workers at the asbestos plant in Aalborg has been examined in previous cohort studies [16]. Raffn et al. found a significantly increased overall incidence (SIR = 1.22, 95% CI 1.12–1.32) of cancer among men employed at this plant compared with all Danish men in the period between 1928 and 1984 [16]. Significant excess risks for cancer among the male workers were found for cancer of the lung, pleura, mediastinum, stomach, penis, and larynx for the group of asbestos cement workers employed between the years 1928 to 1940, with 15 years of latency [16].

The aim of the present study was to examine the risk of all types of cancer, including asbestos-associated cancers and multiple cancers in the same subject, in a cohort of former school children who were exposed to environmental asbestos from the asbestos cement plant in their neighborhood

## 2. Materials and Methods

### 2.1. Population

School children from four primary schools located in the prevailing wind direction at a distance of 100–750 m from the asbestos cement plant near the center of Aalborg city were selected for the study [17]. From the Aalborg City Archives, we retrieved 17,838 seventh grade school records from the four schools on all former pupils born between 1940 and 1970. These pupils were identified through their unique 10-digit personal identification number (CPR number). A CPR number has been assigned to all residents in Denmark by the Danish Civil Registration System (CRS) since 2 April 1968 [18]. Examination of the school records led to exclusion of some pupils due to lack of unique identification (***n*** = 1214), invalid CPR number (***n*** = 139), birth year either before 1940 or after 1970 (***n*** = 1649), and multiple records (***n*** = 2723). Multiple records occurred because some pupils had attended more than one of the four schools. A reference cohort, frequency matched 1:9 on sex and five-year age intervals, was sampled from the CRS. Furthermore, we excluded subjects who had emigrated (nReference = 6), died (nReference = 1), or been diagnosed with cancer (nSchool = 2, nReference = 32) before start of school in the year of their twelfth-year birthday. 

### 2.2. Asbestos Exposure

School cohortees were, per definition, environmentally exposed, not only by attending the schools near the factory, but they also lived in the same neighborhood because Danish children were normally assigned to the school nearest to their place of residence. Measurements of asbestos exposure at the asbestos plant in 1948 performed by the Danish National Institute of Occupational Health indicate that asbestos fiber levels may have been 100–1600 times above the present Danish threshold limit value of 0.5 fiber/mL (9). Asbestos production was relatively low-tech, so we assume that there was some asbestos pollution spilled from the plant to its neighborhood. Furthermore, until 1972, asbestos was transported from the port to the plant in hessian bags, allowing leaks and spills [16]. The reference cohort had no known environmental asbestos exposure.

In order to reconstruct potential additional occupational asbestos exposure, we obtained employment history based on individual records from the Danish Supplementary Pension Fund Register (ATP). The ATP has been registering all wage earners aged 16–66 years, working at least 9 h/week, since 1 April 1964, and the register includes company information and date of start and end of each employment. Furthermore, we obtained information from Statistics Denmark on the classification of companies in which cohortees had been employed. Companies were classified according to a five-digit extended version (DSE77) of the International Standard Industrial Classification of all Economic Activities (ISIC) [19]. To assess occupational asbestos exposure, we used DANJEM based on the Nordic Occupational Cancer Study (NOCCA) job-exposure matrix (JEM) [20]. We have edited the JEM to make it compatible with Danish industry codes (DSE77) and supplemented the JEM with data on specific companies with a known historical use of asbestos. The DANJEM characterizes asbestos exposure by estimates of the prevalence and level of exposure in four periods from 1945 until 1994 [20]. Previously, it has been estimated that in the period until the Danish asbestos ban, approximately 150,000 persons were exposed to occupational asbestos, corresponding to approximately 10% of the working population [21]. We classified a cohortee as ever having been exposed to occupational asbestos if the exposure prevalence in the JEM exceeded 50% in at least one job held in the period from April 1964 until 31 December 1994. By this categorisation, 10.0% of the reference cohort were exposed to occupational asbestos exposure. A subject was grouped as exposed from relatives (household exposure) if a relative was assessed as being exposed to occupational asbestos in this period. Relatives, i.e., spouse, mother, father, siblings, and children under 18 years of age, were identified in the CRS by their CPR number. Their individual employment histories, backdating to 1964, were extracted from the ATP. The index subject’s mother and father were defined as ever exposed to occupational asbestos if exposure took place in the period from when the index subject was born (at the earliest, April 1964) to his or her 18th birthday. Spouses and children of the index person were defined as ever exposed to occupational asbestos if the index subject was 18+ years old and under 18 years old, respectively. An individual from the school cohort was defined as being exposed to environmental asbestos in the absence of their own occupational asbestos exposure and household asbestos exposure. The reference cohort does not have a known asbestos exposure besides the occupational and household asbestos exposures.

### 2.3. Cancer

All cancers diagnosed in Denmark have been registered in the Danish Cancer Registry since 1943. The Danish Cancer Registry has a high degree of completeness and accuracy [22]. By linking the CPR number to the Danish Cancer Registry, data on cancer type and date of diagnosis were retrieved for both the school cohort and the reference cohort. The asbestos cancer diseases in our study were selected on the basis of the IARC’s classification of cancer sites associated with asbestos exposure, including mesothelioma and cancer of the lung, stomach, colon, rectum, larynx and pharynx, and ovaries [8]. The diagnoses were classified according to extended Danish versions of the International Classification of Diseases version ICD-7 (1943–1977), ICD-O (1978–2003), and ICD-10 (2004 and onwards) [22]. Follow-up for cancer began 2 April 1968 (start date for CRS) or 1 August the year the subjects turned 12 years old (seventh grade school start) if born after 2 April 1968. It ended on the date of death, emigration, disappearance, or 31 December 2015, whichever came first.

### 2.4. Tobacco Smoking

Smoking data were available only for some of the lung cancer cases in the Danish Lung Cancer Registry (DLCR), from which we retrieved dichotomized smoking data (ever/never smoker). The DLCR contains information about characteristics in lung cancer patients registered since the year 2000; since November 2013, patient reported outcome measures have been included, as well [23].

### 2.5. Statistics

The chi-square test was used to compare categorical variables between the two groups of the study population. Age medians were analysed using the Wilcoxon–Mann–Whitney test. Person-years at risk for each subject were calculated according to the follow-up period and split into five-year age and calendar time intervals. SIRs with corresponding 95% confidence intervals (95% CIs) were estimated as the ratio between the observed number of cancers in the school cohort and the expected number of cancers in the reference cohort. In order to estimate the risk significance of the various asbestos exposures, the cohorts were stratified into “environmental asbestos exposure”, “household asbestos exposure”, “occupational asbestos exposure”, and “occupational and household asbestos exposure”. Statistical analyses were performed using Stata 15.1 (Stata Corp LLC, College Station, TX, USA). 

The study was performed in accordance with the Helsinki Declaration and approved by the Danish Data Protection Agency (j. no.: 2016-41-4787).

## 3. Results

The final school cohort consisted of 12,111 children who attended one of the four schools nearby the asbestos cement plant (50.3% males and 49.7% females), contributing 592,986 person-years at risk. Table 1 presents the characteristics of the school children and the reference group. The median attained age at the end of follow-up was 62.5 years in the school cohort, with the fewest (3.7%) born in the latest birth period from 1965 to 1970. Significantly more subjects in the school cohort had been occupationally asbestos exposed compared to the reference cohort (16.9% versus 10.0%). 

### 3.1. Cancer Incidence Ratios

Table 2 presents the SIRs for the school cohort and the reference cohort. After exclusion of non-melanoma skin cancers, 1827 individuals in the school cohort were registered with at least one primary cancer, 38 of which were malignant mesothelioma. The SIR for malignant mesothelioma was the highest of all the asbestos-associated cancers (SIR 8.77, 95% CI 6.38–12.05). We found a significantly increased incidence both in the group of all cancer types (SIR 1.07, 95% CI 1.02–1.12) and in the group of all asbestos-associated cancers (SIR 1.14, 95% CI 1.05–1.24) compared with the reference cohort. In a sub-analysis excluding malignant mesothelioma cases, the overall cancer incidence ratio was marginally, but significantly, increased (SIR 1.05, 95% CI 1.00–1.10). However, the SIR for all asbestos-associated cancers did not reach significance when malignant mesothelioma was excluded (SIR 1.07, 95% CI 0.99–1.17). Besides malignant mesothelioma, no other asbestos-associated cancer type was significantly elevated, even though an excess risk of cancer of the colon, larynx, lung, pharynx, and stomach was observed. The incidence of ovarian cancer was significantly lower. In total, 155 individuals (1.28%) in the school cohort and 1263 individuals (1.16%) in the reference cohort were diagnosed with more than one cancer. For those with an asbestos-associated cancer as their first primary cancer, 60 persons (10.8%) from the school cohort had more than one cancer (maximum of two cancers), versus 554 persons (12.8%) in the reference cohort (maximum of four cancers).

### 3.2. Cancer Incidence Rates Related to Type of Exposure

All school cohortees were assumed to have been exposed to environmental asbestos. An analysis was performed dividing the cohort into subgroups according to possible additional asbestos exposure in various combinations (household, occupational, and both occupational and household) (Table 3). When the cohort was divided into additional asbestos exposure subgroups, the SIR for all cancers was only significantly increased in the subgroup of school children who were exposed to both environmental and occupational asbestos, and this significance disappeared in the sub-analysis in which malignant mesothelioma was extracted from the ‘all cancers’ subgroup (SIR 1.11, 95% CI 1.00–1.24). 

Among those with an asbestos-related cancer, the SIR reached significance only in the subgroup combining environmental and occupational asbestos exposure. The SIR remained significant after extracting malignant mesothelioma from ‘all asbestos associated cancers’ (SIR 1.30, 96% CI 1.09–1.55).

Analysis by cancer type showed an increased SIR for malignant mesothelioma in all combinations of asbestos exposure and in those exposed only to environmental asbestos; the SIR was five-fold increased (SIR 5.09, 95% CI 2.82–9.20). 

Cancer of the pharynx was significantly increased in the subgroup of environmental combined with occupational and household asbestos exposure (SIR 4.24, 95% CI 1.59–11.29). The SIR for lung cancer was significantly higher for those school children who had been exposed to both environmental and occupational asbestos (SIR 1.34, 95% CI 1.05–1.72). 

### 3.3. Tobacco-Related Cancers

From the DLCR, we received data on 163 lung cancer cases (69.1%) from the school cohort and 2802 (66.6%) lung cancer cases from the reference cohort. In the DLCR, all lung cancer cases from both cohorts were registered as current or former smokers. 

Asbestos-associated cancers are also tobacco-related, with the exception of malignant mesothelioma and most pathological types of ovarian cancer [24]. In a sub-analysis, the incidence of an asbestos and tobacco-associated cancer (colon, larynx, lung, pharynx, rectum, and stomach) was found to be significantly increased compared with the reference cohort group (SIR 1.11, 95% CI 1.02–1.21).

## 4. Discussion

Our study is the first to investigate the total burden of cancer following primarily environmental exposure to asbestos in childhood. In the present study, which provides follow-up data on cancer for almost 50 years, we investigated the incidence of overall primary cancer; multiple primary cancers; mesotheliomas; and cancers of the lung, larynx, pharynx rectum, stomach, and ovaries in children attending schools and living near a high-volume asbestos cement plant. Furthermore, we assessed risks from additional occupational asbestos exposure and family carry-home asbestos exposures. Overall, the incidence of cancer was increased. This increase was primarily driven by an increased risk of mesothelioma. However, in the subgroup of study participants who had been exposed to occupational asbestos, in addition to childhood environmental asbestos exposure, the risk of asbestos-associated cancers remained significantly elevated even when mesothelioma was excluded from the group.

Several previous studies have established that asbestos exposure is the dominant cause of malignant mesothelioma [25,26]. Malignant mesothelioma occurs even after low-level exposures, and apparently no safe level exists [27]. Only few studies have investigated the effect of childhood asbestos exposure and the risk of mesothelioma [10,28,29,30,31,32,33,34,35,36,37]. However, in a residentially crocidolite asbestos-exposed cohort from Wittenoom, Australia, the mesothelioma mortality rate was lower in those first exposed as children than in those first exposed at >15 years of age [31]. Similar results have been reported in a cohort of asbestos workers in Hong Kong [32]. In contrast, a British case-control study found the odds ratio (OR) to be higher in subjects who were younger than 20 years at first exposure than in subjects aged 30 years or more at first exposure. However, no increased OR was found for those living within one mile of a potential source (asbestos factory, disposal site, shipyard, or power plant) before 30 years of age [33]. In a study from New Caledonia (South Pacific), the risk of mesothelioma was found to be strongly associated with the use of whitewash using tremolite asbestos derived from local outcroppings. All cases’ exposure had begun before the age of 16 [34]. Additionally, in a cohort employed by an Italian asbestos company producing mainly textiles, the standardized mortality ratio of mesothelioma was strongly related to time since first exposure [35]. Finally, in a French study, the effect of the total duration of asbestos exposure was found to decrease when age at first exposure and time since last exposure increased [36]. Accordingly, the majority of previous studies have reported a higher risk of mesothelioma if subjects were exposed in childhood than if they were exposed later in life [28]. This is in line with the significantly increased risk of mesothelioma found in the asbestos-exposed school cohort in the present study. The low number of cases in the two subgroups with additional asbestos exposure from household and from both occupation and household could explain the wide confidence intervals and may also explain the higher risk in those who were also exposed to household asbestos. 

For lung cancer, the risk has been found to increase linearly with the cumulative dose of asbestos exposure [38,39,40]. Studies investigating lung cancer risk have often involved relatively heavy asbestos exposure, and the risk at lower asbestos exposure levels has then been extrapolated from this reference threshold [41]. However, in a recent review and meta-analysis evaluating the association between environmental asbestos exposure and lung cancer, a significant increase in the risk of lung cancer was found for neighbourhood asbestos exposure [42]. We found the SIR for lung cancer to be significantly higher for the school children exposed to both environmental and occupational asbestos (SIR 1.34, 95% CI 1.05–1.72) than for the reference cohort, which suggests that the increase may originate in particular from occupational exposure, since no effect was observed in the groups without occupational exposure. However, a substantially higher frequency of tobacco smoking in the group exposed to occupational asbestos than in the group that was not exposed to occupational asbestos may also contribute to the observed increased lung cancer risk. The majority of lung cancer cases from both cohorts have been registered in the DLCR as former or current smokers, and there is well-established evidence for the additive or even multiplicative effect of asbestos exposure along with smoking on the risk of developing lung cancer [43,44]. However, due to lack of detailed tobacco data, a firm conclusion on this cannot be drawn.

Furthermore, along with tobacco smoking, alcohol drinking is also a risk factor for developing cancers of the pharynx, larynx, colon, and rectum [24,45]. A significantly increased incidence of stomach and laryngeal cancer was reported in the cohort of male workers employed at the Danish asbestos cement plant 1928–1984 [16]. Few other studies examining the correlation between asbestos exposure and laryngeal cancer have been able to determine a causal association between asbestos exposure and laryngeal cancer, and often studies have not accounted for confounding factors [46]. We observed an excess risk of laryngeal cancer, however, the incidence was not significantly increased compared with the reference cohort; nor did we have detailed data on confounding factors. 

Results from a meta-analysis confirm the overall conclusion by the IARC, viz. that exposure to asbestos is associated with a modest, increased risk of stomach cancer [47]. However, in a previous study analyzing mortality from cancer of the lung and stomach in a town with asbestos deposits and in a town with asbestos processing, no significant differences could be attributed to environmental asbestos exposure [48]. This is consistent with the modest, but non-significantly increased, risk for stomach cancer we found in the school cohort. 

The significant reduction in ovarian cancer observed in our study is not seen in previous studies [49]. Furthermore, the significance of our finding disappeared once the cohort was divided into asbestos exposure subgroups, which points to no protective effect of asbestos exposure as far as the incidence of ovarian cancer is concerned.

Few previous studies have investigated the influence of asbestos on developing multiple cancers [12,13]. To our knowledge, no studies have examined the incidence of developing multiple primary cancers after environmental asbestos exposure. The incidence of multiple cancers in the school cohort was similar to that of the reference cohort, both in the overall school cohort and when the cohort was divided into the asbestos exposure subgroups. This limits support for the hypothesis that the carcinogenic effect of asbestos could cause multiple cancers. Further research on this is warranted. 

We acknowledge certain limitations of the present study. First, potential confounders such as smoking, alcohol consumption, and other lifestyle factors may have affected the incidence of certain cancers studied. Our study is registry-based, and the data analyzed did not include these potentially confounding variables. Due to lack of relevant co-carcinogens or confounders, the risk estimates associated with asbestos exposure might be higher than the true estimates. Assessment of occupational exposure and household asbestos exposure by linkage to a JEM may result in non-differential misclassification and may attenuate the SIR towards the null. 

Our findings are based on data from a large population. This, as well as the very long follow-up time and the use of long-established, practically complete and objective, national registers, provides valid and robust results. Thus, for settings similar to Aalborg, we find the external validity of this study to be high.

## 5. Conclusions

Both the incidence of malignant mesothelioma and the overall cancer incidence were significantly increased among our cohort of former school children who attended school and lived near a large asbestos cement plant. Our results confirm the strong association between environmental asbestos exposure and malignant mesothelioma. This study indicates that not only does occupational asbestos exposure pose an increased risk of cancer, but environmental asbestos exposure in childhood may also have deleterious effects. Asbestos remains in use in several countries around the world, and more attention should be paid to prevent the carcinogenic effects of asbestos from environmental, occupational, and household exposure to asbestos. The most effective prevention is to stop the use of asbestos. Further studies are warranted. 

## Figures and Tables

**Table 1 ijerph-19-00268-t001:** Characteristics of the Aalborg School Children Cohort and the sex and age-matched reference cohort.

	School Cohort (*n* = 12,111)		Reference Cohort (*n* = 108,987)		
Characteristics	*n*	(%/range)	*n*	(%/range)	*p*-value
Sex					
Male	6087	(50.3)	54,787	(50.3)	
Female	6024	(49.7)	54,200	(49.7)	
Birth-year					
1940–1944	2409	(19.9)	21,687	(19.9)	
1945–1949	2961	(24.5)	26,645	(24.5)	
1950–1954	2671	(22.1)	24,029	(22.1)	
1955–1959	2166	(17.9)	19,498	(17.9)	
1960–1964	1451	(12.0)	13,047	(12.0)	
1965–1970	453	(3.7)	4081	(3.7)	
Person-years of follow-up	592,986		5,131,278		
Median age at analysis	62.5	(13.5–76.0)	62.2	(12.0–76.0)	0.001
Type of asbestos exposure					0.000
Only environmental asbestos exposure/No known asbestos exposure	8013	(66.2)	83,525	(76.6)	
Occupational asbestos exposure	1761	(14.5)	9685	(8.9)	
Household and occupational asbestos exposure	1916	(15.8)	10,398	(9.5)	
Occupational and household occupational asbestos exposure	287	(2.4)	1203	(1.1)	
No Supplementary Pension Fund Register data	134	(1.1)	4176	(3.8)	

**Table 2 ijerph-19-00268-t002:** Standardized incidence ratios for cancer (1968–2015) among 12111 former school children from Aalborg.

	All School Children		
Cancer Site	O/E	SIR	(95% CI)
All cancers (minus non-melanoma skin cancers) *	1827/1706.82	**1.07**	**(1.02–1.12)**
All asbestos-associated cancers *	580/509.35	**1.14**	**(1.05–1.24)**
Mesothelioma	38/4.33	**8.77**	**(6.38–12.05)**
Lung	236/213.50	1.11	(0.97–1.26)
Larynx	25/17.26	1.45	(0.98–2.14)
Ovary, fallopian tube and broad ligament	33/46.73	**0.71**	**(0.50–0.99)**
Pharynx	40/30.43	1.31	(0.96–1.79)
Stomach	33/28.56	1.16	(0.82–1.63)
Colon incl. rectosigmoideum	119/112.41	1.06	(0.88–1.27)
Rectum	67/69.29	0.96	(0.75–1.21)
Multiple cancers (>1 cancer)	155/143.74	1.08	(0.92–1.26)

* Individuals with at least one cancer; Bold denotes statistically significant results, *p* < 0.05; Abbreviations: CI, confidence interval; E, expected number of cases; O, observed number of cases, SIR, standardized incidence ratio.

**Table 3 ijerph-19-00268-t003:** Standardized incidence ratios (SIR) for selected cancers among those in the school cohort exposed only to environmental asbestos and the subgroups also exposed to household or/and occupational asbestos exposure.

	Environmental Asbestos Exposure			Household Asbestos Exposure			Occupational Asbestos Exposure			Occupational and Household Asbestos Exposure		
Cancer Site	O	SIR	(95% CI)	O	SIR	(95% CI)	O	SIR	(95% CI)	O	SIR	(95% CI)
All cancers (minus non-melanoma skin cancers) *	1180	1.05	(0.99–1.11)	257	0.95	(0.84–1.08)	**335**	**1.18**	**(1.06–1.31)**	49	1.12	(0.84–1.48)
All asbestos associated cancers *	339	1.03	(0.93–1.15)	73	0.96	(0.76–1.21)	**144**	**1.47**	**(1.25–1.74)**	21	1.33	(0.87–2.04)
Malignant mesothelioma	**11**	**5.09**	**(2.82–9.20)**	**4**	**22.77**	**(8.55–60.67)**	**20**	**8.67**	**(5.60–13.44)**	**3**	**10.38**	**(3.35–32.19)**
Lung, bronchus and trachea	135	1.01	(0.85–1.20)	32	0.93	(0.66–1.32)	**62**	**1.34**	**(1.05–1.72)**	7	0.88	(0.42–1.85)
Larynx	14	1.35	(0.80–2.28)	3	1.87	(0.60–5.79)	8	1.30	(0.65–2.60)	0	-	-
Ovary, fallopian tube and broad ligament	27	0.81	(0.56–1.19)	3	0.33	(0.11–1.02)	2	2.17	(0.54–8.67)	0	-	-
Pharynx	17	0.90	(0.56–1.44)	6	2.01	(0.90–4.47)	13	1.58	(0.91–2.71)	**4**	**4.24**	**(1.59–11.29)**
Stomach	22	1.19	(0.79–1.81)	2	0.55	(0.14–2.18)	8	1.38	(0.69–2.77)	1	1.20	(0.17–8.51)
Colon incl. rectosigmoideum	79	1.06	(0.85–1.32)	16	1.06	(0.65–1.73)	20	0.99	(0.64–1.54)	3	1.00	(0.32–3.09)
Rectum	40	0.87	(0.64–1.18)	9	0.83	(0.43–1.59)	14	1.27	(0.75–2.14)	3	1.26	(0.41–3.90)
Multiple cancers (>1 cancer)	100	0.96	(0.79–1.17)	22	1.00	(0.66–1.52)	30	1.20	(0.84–1.72)	3	0.63	(0.20–1.94)

Abbreviations: CI, confidence interval; O, observed number of cases, SIR, standardized incidence ratio; * Individuals with at least one cancer; 6 of the former school children with cancer (3 of them with an asbestos associated cancer) have no data on employments; Bold denotes statistically significant results.

## Data Availability

Not applicable.

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
