# Peer review of "Cancer Incidence and Risk of Multiple Cancers after Environmental Asbestos Exposure in Childhood—A Long-Term Register-Based Cohort Study"

_ijerph, 2021, doi:10.3390/ijerph19010268_

Round 1

Reviewer 1 Report

This is the third time that I am being asked to review this work.

It is a valuable study and revisions done by the authors are acceptable.

The authors are strongly recommended to provide a reasonable Consent Statement and Data Availability Statement. Especially the Data Availability Statement is extremely important for further studies.

Except that I have no further comments.

Author Response

Thank you for your comments.

Informed Consent Statement: According to Danish law, register-based studies do not need approval by ethical and scientific committees, nor informed consent. 

Reviewer 2 Report

Modest alterations were performed, but maintained the significance of the work.

Author Response

Thank you for your comments

This manuscript is a resubmission of an earlier submission. The following is a list of the peer review reports and author responses from that submission.

Round 1

Reviewer 1 Report

This paper is a valuable report of a cohort study on cancer incidence because of asbestos exposure in childhood.

The introduction is short, sharp, and informative. The study procedure is sound and results are valuable for further practical applications in risk assessment.

I only have one question: what were the exposure routes (oral, dermal, or inhalation)? It will be great if the authors may discuss it in more detail.

If possible and applicable, I also wish to invite the authors to present their raw data either as supplementary material or in a publicly accessible repository. In follow, please do provide a "Data Availability Statement" since it can be very helpful for further studies.

Author Response

Dear Reviewer 1. 

Thank you for your review of our paper. 

As you are aware of the study is register based without any further information of routes of exposure. We strongly believe however, that the children were primary exposed by inhalation and to some extent by ingestion. A sentence to clarify this has been added to the introduction. Due to lack of relevant information further discussion has not been added to the manuscript.  

Unfortunately it is not possible to present the raw data as supplementary material or in a publicly accessible repository.

Reviewer 2 Report

In the manuscript ijerph-1418583, the authors aimed to associate environmental asbestos exposure and cancer risk development.

I've found a few points that in my point of view should be clarified:

How did the authors divide those children who were at occupational exposure from those who were not exposed? Please clarify this, since you mentioned that the reference group was not exposed to asbestos (at least with unknown exposure)

It is difficult to ascribe the development of those diseases only to asbestos exposure. How can this factor (asbestos exposure) be corrected?

It would be interesting to understand how much those confounders (smoking, alcohol, and other relevant lifestyle factors) contribute negatively or positively (exacerbation) to the disease's development.

References should be revised and formatted accordingly.

Author Response

Dear reviewer 2.

Thank you for your review of our paper. 

How did the authors divide those children who were at occupational exposure from those who were not exposed? We did so by means of data from Danish Supplementary Pension Fund Register (ATP), Statistics Denmark, and DENJEM assessing occupational asbestos exposure. Due to the unique identification number CPR we were able to calculate the occupational asbestos exposure for each individual. By this categorization, 10.0% of the reference cohort were exposed to occupational asbestos exposure. The reference cohort does not have a known exposure besides the occupational and household asbestos exposure. Section 2.2 Asbestos exposure describe in more detail the steps in the calculation process.  

It is difficult to ascribe the development of those diseases only to asbestos exposure. How can this factor (asbestos exposure) be corrected?

We agree that it is difficult to ascribe the development of certain cancer to asbestos exposure. The epidemiological evidence for a positive association between asbestos exposure and cancer diseases  has been assessed by the IARC, which we used as a reference in the introduction. For mesothelioma it has been established that asbestos exposure is the dominant cause of malignant mesothelioma and for lung cancer the risk has been found to increase linearly with the cumulative dose of asbestos exposure. The epidemiological evidence for and a positive association between asbestos exposure and the remaining cancers are more limited.

The study is a register based historical cohort study with no information of smoking habits, alcohol consumption and other important lifestyle events. We totally agree with the reviewer in his arguments, but due to the study design we are unable to do these crucial analyses. We have therefore added to the manuscript that further studies are warranted to underline that our findings are based on crude analysis with no confounder control.

Reviewer 3 Report

This article shows a link between environmental asbestos exposure and malignant mesothelioma, confirming that not only occupational asbestos exposure poses an increased risk of cancer, but also environmental asbestos exposure in childhood.

The study is not original but it will be useful for authorities of countries where asbestos remains in use. Moreover, it is relevant from the authors to be cautious in their conclusions by acknowledging certain limitations of their study and by specifying that further research is warranted. 

Author Response

Dear reviewer 3.

Thank you for your review of our paper and thank you for your comments and suggestion to specify the need for further research on this subject. We did list limitations in the study, but we certainly acknowledge that further research is warranted. To underline this  we have added the sentence "Further studies are warranted." as the very last sentence.